# Effectiveness of COVID-19 Vaccination on Reduction of Hospitalizations and Deaths in Elderly Patients in Rio Grande do Norte, Brazil

**DOI:** 10.3390/ijerph192113902

**Published:** 2022-10-26

**Authors:** Ana Isabela L. Sales-Moioli, Leonardo J. Galvão-Lima, Talita K. B. Pinto, Pablo H. Cardoso, Rodrigo D. Silva, Felipe Fernandes, Ingridy M. P. Barbalho, Fernando L. O. Farias, Nicolas V. R. Veras, Gustavo F. Souza, Agnaldo S. Cruz, Ion G. M. Andrade, Lúcio Gama, Ricardo A. M. Valentim

**Affiliations:** 1Laboratory of Technological Innovation in Health (LAIS), Hospital Universitário Onofre Lopes, Federal University of Rio Grande do Norte (UFRN), Natal 59012-300, RN, Brazil; 2Rio Grande do Norte School of Public Health (ESPRN), Natal 59015-350, RN, Brazil; 3Department of Molecular and Comparative Biology, Johns Hopkins University School of Medicine, Baltimore, MD 21205, USA; 4Vaccine Research Center, National Institute of Allergy and Infectious Diseases (NIAID), National Institutes of Health (NIH), Bethesda, MD 20892, USA

**Keywords:** COVID-19, immunization, effectiveness, hospitalization

## Abstract

Since the COVID-19 pandemic emerged, vaccination has been the core strategy to mitigate the spread of SARS-CoV-2 in humans. This paper analyzes the impact of COVID-19 vaccination on hospitalizations and deaths in the state of Rio Grande do Norte, Brazil. We analyzed data from 23,516 hospitalized COVID-19 patients diagnosed between April 2020 and August 2021. We excluded the data from patients hospitalized through direct occupancy, unknown outcomes, and unconfirmed COVID-19 cases, resulting in data from 12,635 patients cross-referenced with the immunization status during hospitalization. Our results indicated that administering at least one dose of the immunizers was sufficient to significantly reduce the occurrence of moderate and severe COVID-19 cases among patients under 59 years. Considering the partially or fully immunized patients, the mean age is similar between the analyzed groups, despite the occurrence of comorbidities and higher than that observed among not immunized patients. Thus, immunized patients present lower Unified Score for Prioritization (USP) levels when diagnosed with COVID-19. Our data suggest that COVID-19 vaccination significantly reduced the hospitalization and death of elderly patients (60+ years) after administration of at least one dose. Comorbidities do not change the mean age of moderate/severe COVID-19 cases and the days required for the hospitalization of these patients.

## 1. Introduction

As soon as the first COVID-19 cases emerged and the World Health Organization (WHO) declared the pandemic in March 2020, research groups and pharmaceutical companies worldwide responded by repurposing potentially effective drugs to treat moderate to severe cases and developing effective vaccines to prevent transmission among vulnerable individuals [1,2,3,4]. Currently, over 7000 studies related to COVID-19 are registered at *ClinicalTrials.gov*, (accessed on 22 April 2022), of which 1386 are associated with developing vaccine candidates or evaluating the potential cross-immunity to COVID-19 after BCG vaccination [5]. Given the alarming number of new infections and COVID-19 lethality globally, the Food and Drug Administration (FDA) granted the first emergency use authorization for the Pfizer/BioNTech vaccine in December 2020 after evaluating data from Phase 3 clinical results. At that point, the agency allowed its administration to patients aged 16 years and older [6]. Over the following months, other regulatory agencies worldwide granted emergency use authorization or regular approval to this and other COVID-19 vaccines.

Given the scarcity of vaccines and other supplies necessary to manage the COVID-19 crisis, especially in low- and middle-income countries, Brazil faced its worst pandemic months (with the highest number of deaths) between March and June 2021. At that time, Brazil became the world's epicenter for new daily infections and deaths, several hospitals collapsed, and massive viral transmissibility allowed the development of new strains, such as the gamma variant [7,8,9,10]. Nonetheless, even in the face of this utterly adverse scenario, the progressive advance of large-scale vaccination resulted in a drastic reduction of new daily cases and deaths observed between July and August 2021. It also minimized subsequent waves of infection, notably caused by delta and omicron variants [11,12,13,14,15]. As a result, the Brazilian COVID-19 scenario presents a sustained reduction of new daily cases and deaths, comparable to April/May 2020 when the first cases were detected [16].

During the initial acquisition of vaccines offered by companies, miscoordination led to the adoption of different immunizers throughout the vaccination process, resulting in an occasional mix-and-match strategy using the BNT162b2 (Pfizer/BioNTech), CoronaVac (Sinovac Biotech), ChAdOx1 nCoV-19 (Oxford/AstraZeneca), and Ad26.COV2.S (Janssen) vaccines, becoming evident a few months later as an efficient strategy to amplify the immune response during COVID-19 vaccination; now also encouraged in several countries [17,18,19,20,21,22,23,24].

At the beginning of the Brazilian COVID-19 vaccination campaign, the Ministry of Health (MoH) recommended that specific groups be prioritized during the initial campaign (i.e., health professionals, essential workers, residents of long-term care institutions, and the elderly ≥ 80 years). However, with the large-scale availability of new doses, the Brazilian MoH extended the campaign to people with pre-existing medical conditions and younger age groups in descending order [25].

Considering the expressive societal adherence to COVID-19 vaccines in Brazil, this study aimed to evaluate the impact of the immunization process on moderate or severe COVID-19 cases in elderly patients and explore how those patients performed considering their vaccination status at the time of admission in the state of Rio Grande do Norte (RN), Brazil. Crossing data from the vaccination and hospitalization systems, we explored the impact of vaccination on the decline in new infections; hospitalizations (ICU or clinical units); length of hospitalization; the severity of cases according to the Unified Score for Prioritization (USP), which combines the parameters described in the quick Sequential Organ Failure Assessment (qSOFA), the Charlson Comorbidity Index (CCI), the Clinical Frailty Scale (CFS), and The Karnofsky Performance Status scores; comorbidities, and outcome (discharge or death) after hospitalization [26,27].

## 2. Materials and Methods

### 2.1. Caseload and Data Acquisition

Among the actions coordinated to face the COVID-19 pandemic in RN, Brazil, our group at the Laboratory of Technological Innovation in Health (LAIS/UFRN) partnered with the Secretary of Public Health of Rio Grande do Norte (SESAP/RN) and the MoH to develop a technology ecosystem for health services [28]. It included a regulatory system for access to clinical and intensive care unit (ICU) beds for COVID-19 patients (RegulaRN) and the RN + Vacina, a vaccination system requiring registration for the general population [29]. Taken together, RegulaRN and RN + Vacina fully centralized the process of hospital bed management (ICU or clinical units) and tracking vaccination progress across the state, which allowed for timely interventions and fostered resilience in the Brazilian National Health System (SUS) at a local level, thus averting the RN health system collapse during such a severe sanitary crisis [28].

Hence, considering the acquisition and crossing of the dataset of hospitalized patients (ICU or clinical units, retrieved from RegulaRN) with the vaccination data (obtained from the RN + Vacina dataset), we analyzed data from 23,516 patients hospitalized due to COVID-19 between April 2020 and August 2021. The Secretary of Public Health of Rio Grande do Norte (SESAP/RN) previously anonymized data and its analysis was authorized after being publicly available in the repository. Therefore, we excluded from the analysis data (a) patients hospitalized through direct occupancy (not managed through the system), (b) patients with unknown outcomes (e.g., those transferred to other health centers), and (c) hospitalized patients with an unconfirmed case of COVID-19. Based on such criteria, data from 12,635 patients hospitalized due to COVID-19 (moderate or severe cases) were analyzed and cross-referenced with vaccination data. Considering the hospitalization and lethal outcomes observed in elderly patients, we assessed the immunization process's effectiveness in reducing moderate or severe COVID-19 cases in elderly patients in Rio Grande do Norte (RN), Brazil. Furthermore, considering the immunization status, we classified those hospitalized patients into three independent groups: not immunized (NI), who had not received any dose of available vaccines; partially vaccinated (D1), who received at least one dose for more than 21 days at the time of hospitalization; and fully immunized (D2), those who had taken both doses more than 21 days ago at the time of admission. This timeframe was adopted considering the seroconversion process and the induction of robust humoral and cellular immune responses after each dose [30,31,32,33].

### 2.2. Data Dictionary, Processing of Information, and Interpretation of Results

Each hospitalized patient with a previous immunization record had 46 initial features or attributes that resulted in 16 features after cleaning to extract the summarized data. The complete dataset “covid19_rn-br.csv” was previously anonymized and subjected to peer review to improve the rigor of evaluation and information consistency. The complete dataset presented in this study is available in Zenodo at doi.org/10.5281/zenodo.7249604.

To conduct the analysis, we established several criteria and performed a pipeline (workflow) using Python 3.7.12, applied to the data processing, comprising of three steps: (i) data cleaning/denoising, (ii) feature extraction, (iii) and feature selection. In the first step, we used a screening tool to exclude patients with an unconfirmed COVID-19 diagnosis from the analysis, i.e., suspected, discarded, and unreported cases. Moreover, we excluded from the dataset patients admitted directly by the state's health care network (which frequently does not present complete medical records in the RegulaRN system) and those patients with unknown outcomes (discharge or death) at the end of the hospitalization period.

During feature extraction, new fields in a database were created based on the time required for effective immunity after vaccination (for instance, 21 days after each shot). This feature rendered the following labels: “not immunized” (NI) for those who received a first dose (D1) or not but satisfied the restriction of Equation 1; “partially immunized” for those patients who received only the D1 satisfying the restriction of Equation 1; and “fully immunized,” for those who received their first (D1) and second dose (D2) satisfying the restriction of Equation (1) (as summarized in Equation (1)).
(1)$$date_{hospitalization}-date_{vaccination}\geq 21\;days.$$

Binary features were also established during step (ii) to identify age, whether individuals had comorbidities or not, or if they were health professionals, respectively. During feature selection, only attributes of interest to this study were included in the dataset, whereas attributes that could potentially identify individuals were excluded. In addition, to avoid bias during the data analysis, we explored the period from June to August 2021, in which there was widespread vaccination with two doses for the 60+ years old state population and analyzed the impact of the vaccination process in reducing deaths and hospitalizations among partially (D1) or fully immunized (D2) when compared to those who had not received any dose of the immunizing agent (NI).

### 2.3. Statistical Analysis

All data presented in this study were analyzed using GraphPad Prism 9.4.1 software. The statistical analysis was performed using the Kruskal–Wallis test for group comparison and Dunn’s test to perform multiple comparisons [34,35]. The results are shown as the mean ± standard deviation. Comparisons between columns were considered statistically significant when *p* < 0.05 (* *p* < 0.05; ** *p* < 0.005; *** *p* < 0.001). In addition, we performed an exploratory data analysis using Python programming language. We plotted the data using “matplotlib” and “seaborn” graphic libraries, widely used to create static, animated, and interactive visualizations and high-level interfaces for drawing attractive and informative statistical graphics [36,37].

## 3. Results

### 3.1. Age and Immunization Status, but Not Comorbidities, Are Associated with the Development of Moderate and Severe COVID-19 Cases

According to Brazilian MoH, the COVID-19 vaccination strategy was carried out according to their status of exposition to infection (e.g., health professionals), patients who presented previous medical conditions, and elderly patients in descending order of age considering the vaccine availability. According to RN + Vacina data, until August 2021, the vaccination process reached the 60+ years old state population (*n* = 461.857), resulting in 106.26% (*n* = 490.799) of these public partially (D1) and 98,18% (*n* = 453.494) fully immunized (D2) against COVID-19.

To evaluate the impact of the immunization process during the COVID-19 pandemic, we analyzed 23,516 medical records from COVID-19-related patients hospitalized in ICU or clinical beds from April 2020 to August 2021. Considering only those patients who were adequately managed through the regulation system, with a known outcome (discharge or death) after hospitalization and had a final diagnosis confirming the COVID-19 infection, we examined data from 12,635 patients that required hospitalization due to moderate/severe COVID-19, which resulted in 3949 deaths (31.25%) related to COVID-19 during the analyzed period. In addition, these data were cross-referenced with the vaccination system and reported the immunization status of each patient during the hospitalization. Figure 1 presents a general overview of all analyzed patients' age, comorbidities, outcomes, and immunization statuses.

Our data indicated that NI individuals corresponded for 92% of hospitalizations, whereas partially (D1) or fully immunized (D2) were responsible for 5% and 3% of all analyzed hospitalizations, respectively (Figure 1A). Considering all the discharged patients (*n* = 8686; 68.74% of all patients), we observed that 5514 patients (43.64%) had no comorbidities with a mean age of 50 years, while 3172 patients (25.10%) with one or more comorbidities present a slightly older population with a mean age of 62 years. Among the lethal cases (*n* = 3949; 31.25%), half of the patients (*n* = 1975; 15.63% from all patients) had no comorbidities registered during their hospitalization, and those deaths whose patients had at least one comorbidity presented a mean age of 70 years (Figure 1B).

Among the discharged outcome, 8078 patients (93%) were not immunized (NI) during their hospitalization, of whom 65% (*n* = 5251) had no comorbidity with a mean age of 50 years (vs. mean age of 60 years among those NI with comorbidities), while partially (D1) or fully (D2) immunized discharged patients were equally distributed according to comorbidity criteria. They present a mean age of 70 years. In addition, no case of hospitalization or death was observed among patients under 50 years who received at least one dose of the immunizers, regardless of the manufacturer (Figure 1C). In parallel, 3519 of all deaths (89.11%) occurred in NI individuals and the presence of one or more comorbidities does not significantly change the mean age among groups according to their immunization status. However, age and immunization status appear to be critical variables associated with lethal outcomes, notably when we observed that NI patients are younger (mean 60 years) than partially (mean 71 years) or fully immunized (77 years), independent of the occurrence of comorbidities (Figure 1D). Figure 2 presents the word cloud of the main comorbidities reported among moderate and severe COVID-19 hospitalizations.

Those results suggest that administering at least one dose of the immunizers was sufficient to significantly reduce the occurrence of moderate and severe COVID-19 cases among patients under 59 years. Furthermore, considering the partially (D1) or fully (D2) immunized patients, the mean age is similar between the analyzed groups, despite the occurrence of comorbidities; it is also higher than that observed among NI patients, suggesting that the administration of at least one dose of vaccine is critical to reducing the moderate or severe COVID-19-related cases.

### 3.2. Vaccination Changes the Profile of Hospitalizations and Deaths Related to COVID-19

Considering that the vaccination process was not fully available to all patients during the study, and to avoid bias during our analysis, we carried out a comparative analysis of pre-vaccination (June to August 2020, pre-V) and post-vaccination (June to August 2021, post-v), in which the immunizers were available to all the 60+ years old population and evaluated the occurrence of hospitalization and deaths among these groups.

During the post-V period, most COVID-19 hospitalizations were among NI individuals, corresponding to 78% (*n* = 1590) of all hospitalizations. The analysis of partially (D1) or fully immunized (D2) patients indicated that all patients who developed moderate or severe COVID-19 in this period were 59+ years or older (mean age 67 years or 77.30 years, respectively), in contrast to those NI patients (mean age 47 years), corroborating that at least one dose of immunizers was sufficient to significantly reduce the development of moderate and severe COVID-19 cases (Figure 3A). Similar findings were also observed by analyzing the mean age of groups, considering the occurrence of comorbidities and the outcome (discharge, Figure 3B; or death, Figure 3C).

In the pre-V period, 2288 hospitalizations occurred, of which 68% (*n* = 1554) resulted in discharge and 32% (*n* = 734) in death outcomes. As for the post-V period, there were 2029 admissions due to COVID-19, of which 74% (*n* = 1506) resulted in discharge and 32% (*n* = 523) in death, with a significant number of cases among NI patients (Figure 3D,E). In addition, our results indicate that all moderate/severe COVID-19 cases required a similar period of hospitalization (mean 8.4 days) until the discharge or a lethal outcome. Interestingly, despite the immunization process reducing the number of moderate/severe COVID-19 hospitalization in partially (D1) or fully immunized (D2) patients, a similar period of hospitalization was required for these patients, independent of the existence of previous medical conditions. These results indicate that the immunization process had a critical role in reducing new moderate/severe COVID-19 cases but did not change the mean age of hospitalized patients and the mean hospitalization period of patients with or without comorbidities.

### 3.3. Partially (D1) or Fully Immunized (D2) Patients Present Lower Levels of Unified Score for Prioritization (USP) When Developing Moderate or Severe COVID-19 Cases

Considering the similar period of hospitalization required for the treatment of moderate/severe COVID-19 patients, we examined the severity of cases in NI and partially (D1) or fully (D2) immunized patients hospitalized between June to August/2021, according to the USP. This parameter combined different scales (Sequential Organ Failure Assessment (qSOFA), the Charlson Comorbidity Index (CCI), the Clinical Frailty Scale (CFS), and The Karnofsky Performance Status scores) and was widely used to measure the severity of COVID-19 and prioritize the patient care [26,27].

Partially (D1) or fully immunized (D2) patients present lower USP levels, independent of pre-existing medical conditions, indicating that those patients that had at least one dose under 59 years do not require hospitalization and do present a better prognosis when compared with those NI patients (Figure 4A,B). The mean age of NI discharged patients was 42.78 years old, while partially (D1) or fully immunized (D2) patients presented mean ages of 67.38 and 71.65 years old, respectively. As summarized in Table 1, Table 2, Table 3 and Table 4, discharged patients without comorbidities were hospitalized during this period; 85.04% were NI individuals with lower USP levels (2 or 3). In comparison, partially (D1) or fully immunized (D2) individuals represented 3.24% and 4.08% of these individuals, respectively.

Considering only the patients with a lethal outcome and with any comorbidity (*n* = 255), we observed that NI patients with lower USP levels (2 or 3) represent 36.86% with a mean age of 54.62 years old, while partially (D1, mean age 67.8 years) or fully immunized (D2, mean age 76.5 years) patients represent 7.84% and 8.62% among those lethal cases with comorbidities, respectively (Figure 4C,D). Considering the lethal cases in individuals without comorbidities (*n* = 268), 71.31% of those occurred in NI individuals with lower USP levels (2 or 3) with a mean age of 46.15 years, while partially (D1, mean age 68.1 years) or fully immunized (D2, mean age 77.58 years) were responsible for 4.84% and 6.7% of the deaths among these individuals, respectively.

## 4. Discussion

In Brazil, the first COVID-19 vaccines available to the population were the CoronaVac, produced by Sinovac (China) and Butantan Institute (Brazil), and the Oxford vaccine, produced by AstraZeneca. In January 2021, the Ministry of Health confirmed the delivery of 6 million doses of the CoronaVac Sinovac/Butantan vaccine for all states and the Federal District, thus marking the beginning of vaccination in almost all states [38]. In the first vaccine shipment received by the RN state in January 2021, the SESAP/RN defined health workers, elderly residents in long-term care institutions, and vaccinators as the priority group in the so-called phase 1, followed by the elderly aged 75 or over. In phase 2, older adults aged 60 years or older were included. In phase 3, the people with comorbidities were included. Afterward, in June 2021, vaccination continued to advance in the population in descending order from the age group to the population over 12 years old. The first doses of the CoronaVac/Sinovac/Butantan vaccine were administered to the elderly on 20 January 2021 and the first doses of Oxford/AstraZeneca were administered to the elderly on 27 January 2021.

According to the Brazilian Institute of Geography and Statistics (IBGE), the Rio Grande do Norte state presented a 3,560,903 population, of which elderly patients (60+ years old) represent 9.48% of the total [39]. Considering the end of August 2021, 62.82% of this population was vaccinated with at least one dose, while 24.90% was immunized with two doses. Considering only the elderly population (479,536 habitants), at the end of August 2021, 98% of this population was vaccinated with one dose, and 91.94% of the elderly were immunized with two doses. Although elderly patients still represent a high number of hospitalizations observed in the post-V period (June to August 2021), this number dropped by half compared to the pre-vaccination period analyzed (June to August 2020). In addition, the death rate also decreased in the elderly population after immunization. In the period analyzed in this study, deaths in the elderly correspond to 49.7% of the total. In contrast, during the pre-vaccination period, deaths in the elderly corresponded to 74.5% of the total.

In 2020, the RN state had a fatality rate of COVID-19 of 3.7%. In the post-V period, the fatality rate dropped to 2%. As for specific groups, it is noted that the lethality in the elderly group had a significant drop, falling from 17.14% to 8.26%. The same can be observed in the other groups. In the adult group, the lethality rate dropped by almost half, from 1.27% in 2020 to 0.85% in these three months analyzed, and in the younger group it dropped even more, from 0.43% it gone to 0.15%; that is, there was a decrease of almost three times the fatality rate in the younger group.

These data suggest that aging is a critical feature of moderate and severe COVID-19 cases independent of comorbidities. Furthermore, that vaccination is the main factor in preventing death from COVID-19 infection, notably in elderly patients. The RegulaRN system's data also support these results. Since the beginning of the Rio Grande do Norte vaccination process, it has been possible to observe a reduction in the percentage of critical bed occupancy for 60+ years of patients. These numbers, which have remained very high since the beginning of the pandemic, reached an occupancy rate of 100% of mandatory beds in July 2021, reducing gradually until reaching a stability of 10% occupation that is observed currently.

Despite the occurrence of SARS-CoV-2 variants and the delay in starting the massive vaccination process in Brazil, we are facing a period of stability with low levels of new cases and daily deaths. The initial strategy of herd immunization, encouraged by the federal and several local governments, was proved as an error that resulted in several new cases and deaths within two years of the pandemic. This strategy, associated with the use of proven ineffective drugs and the anti-vaccine movement, also contributed to the rapid spread of COVID-19, the development of new viral strains in Brazil, and the occurrence of three different waves of infection after the dissemination of SARS-CoV-2 variants [40,41,42].

As a limitation of the current study, we did not perform a direct comparison between immunizers in reducing hospitalization and deaths to avoid a predilection for a specific immunizing agent based on its performance in the evaluated population. However, this analysis can be performed using the anonymized dataset available in Zenodo (doi.org/10.5281/zenodo.7249604). Considering that all four immunizers evaluated in this study are approved by WHO and widely administered worldwide, the comparison between the immunizers may represent a relevant aspect for future studies that may help to improve the vaccination process and the development of new vaccines. Along with the immunization process, the Brazilian Ministry of Health made four different vaccine platforms available for the general population, resulting in an occasional mix-and-match strategy using the BNT162b2 (Pfizer/BioNTech), CoronaVac (Sinovac Biotech), ChAdOx1 nCoV-19 (Oxford/AstraZeneca), and Ad26.COV2.S (Janssen). Recently, Clemens and colleagues evidenced the efficacy of the ChAdOx1 nCoV-19 (Oxford/AstraZeneca) vaccine against the viral strains circulating in Brazil. In parallel, Cerqueira-Silva and colleagues explored the effectiveness of these immunizers in individuals with previous SARS-CoV-2 infection [13,43].

In addition, all viral strains identified in this period (including alpha, beta, gamma, and delta) were associated with an increase in transmissibility worldwide but not directly associated with a significant increase in lethality (except when there was a saturation of healthcare services and not directly caused by the increase of viral pathogenesis). Recently, Brizzi and colleagues evidenced that the geographic and temporal fluctuations in Brazil's COVID-19 in-hospital lethality rates between January 2020 and July 2021 were primarily associated with geographic inequities and shortages in healthcare capacity [44], which is also corroborated by the RegulaRN public database [29]. Taken together, our data and the data from worldwide lead us to the unequivocal conclusion that the vaccination process was primarily responsible for reducing new cases and deaths, despite new viral strains in this period.

During this period, the Brazilian health system proved resilient, capable of overcoming this unprecedented challenge and learning along this process. In addition, the strategy to adopt four immunizers (based on different platforms) and the large-scale adhesion of vaccination across the country may help to explain why the subsequent COVID-19 variants did not cause as many lethal cases in Brazil as was observed in other countries in the same period. We strongly encourage new studies to understand the role of the massive vaccination process using different platforms to control COVID-19 spread and how this strategy may help prevent possible outbreaks of this disease in other countries.

## 5. Conclusions

Our data suggest that COVID-19 vaccination significantly reduced the hospitalization and death of elderly patients after administering at least one dose. In addition, our data also demonstrated that comorbidities do not change the mean age of moderate/severe COVID-19 cases and the mean of days required for the hospitalization of these patients. Despite our current advances in understanding COVID-19 dissemination and management, we still need to understand how the adoption of four different platforms of immunization may improve the immune response against SARS-CoV-2 and avoid the occurrence of new outbreaks across the world.

## Figures and Tables

**Figure 1 ijerph-19-13902-f001:**
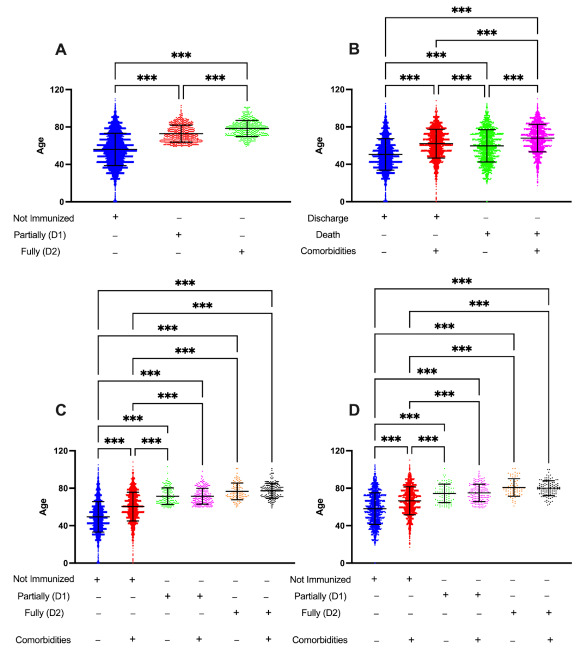
General overview of all analyzed patients according to their age, comorbidities, outcomes, and immunization status. Considering the hospitalizations of moderate and severe COVID-19 cases between April 2020 and August 2021, we analyzed data from 23,516 patients and excluded those hospitalized through direct occupancy, with unknown outcomes, or patients with an unconfirmed COVID-19 case. Based on these criteria, data from 12,635 hospitalized patients were analyzed and cross-referenced with vaccination data. (**A**) Age of hospitalized patients according to their immunization status; (**B**) Age distribution of hospitalized patients according to their outcome and comorbidities; (**C**) Age distribution of discharged patients considering their immunization status and existence of comorbidities; (**D**) Age distribution of lethal cases considering their immunization status and existence of comorbidities. The statistical analysis was performed using the Kruskal–Wallis test for group comparison and Dunn’s test to perform multiple comparisons. The results are shown as the mean ± standard deviation (*** *p* < 0.001).

**Figure 2 ijerph-19-13902-f002:**
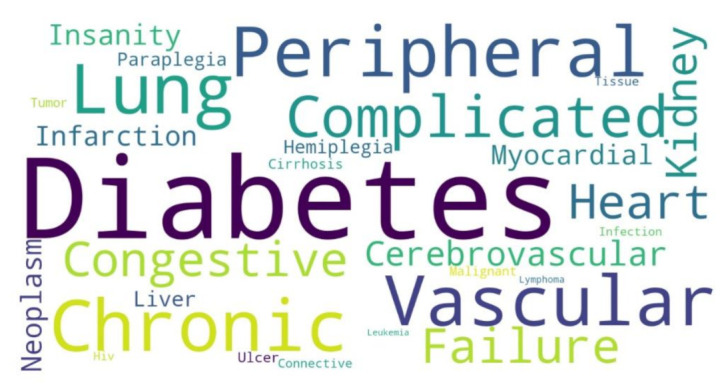
Word cloud of the main comorbidities reported among moderate and severe COVID-19 hospitalizations. Based on RegulaRN data, we identify and analyze the main comorbidities reported on the medical records of each patient. Every comorbidity reported was translated from Portuguese to English and inserted into the word cloud library in Python to generate the word cloud.

**Figure 3 ijerph-19-13902-f003:**
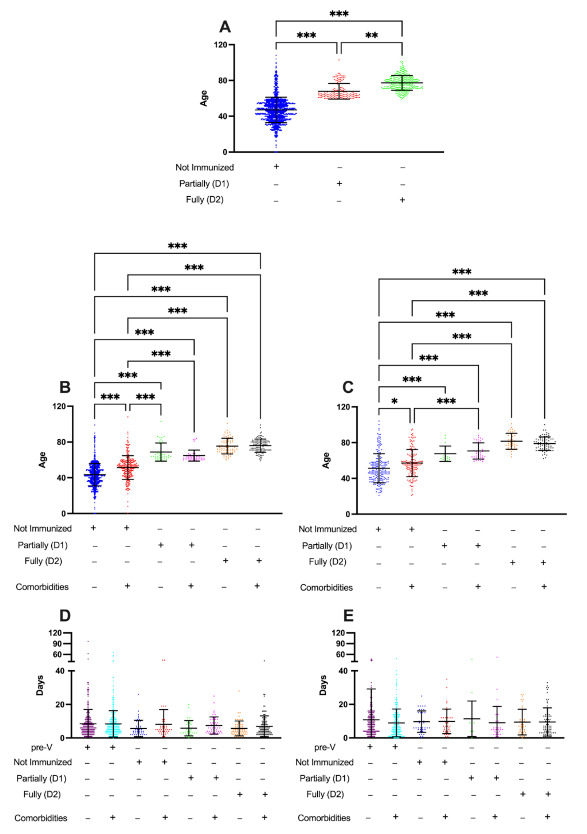
Profile of discharged patients and lethal outcomes related to COVID-19 before and after vaccination. Considering the pre-vaccination (June to August 2020, pre-V) and post-vaccination (June to August 2021, post-v) period, we analyzed the impact of immunization on the profile of hospitalizations and deaths of NI; partially (D1) or fully immunized (D2) patients. (**A**) Hospitalizations related to COVID-19 during the post-V period; (**B**) Profile of discharged patients considering their immunization status and existence of comorbidities; (**C**) Profile of lethal cases considering their immunization status and comorbidities; (**D**) Days required since admission to hospitalization discharge of moderate/severe COVID-19 patients considering their immunization status and the existence of comorbidities; (**E**) Days required since admission to lethal outcome of moderate/severe COVID-19 patients considering their immunization status and the existence of comorbidities. The statistical analysis was performed using the Kruskal–Wallis test for group comparison and Dunn’s test to perform multiple comparisons. The results are shown as the mean ± standard deviation (* *p* < 0.05; ** *p* < 0.005; *** *p* < 0.001).

**Figure 4 ijerph-19-13902-f004:**
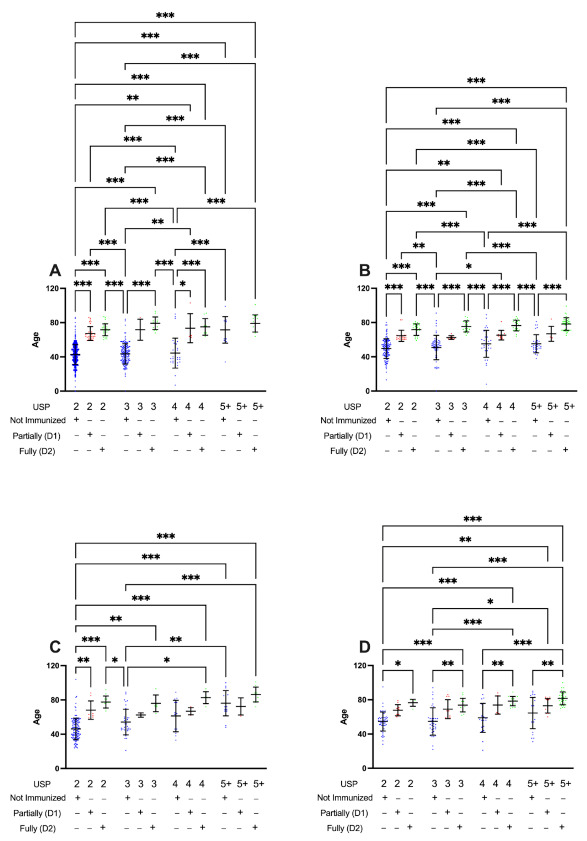
Profile of hospitalization post-vaccination and their influence on the Unified Score for Prioritization (USP). Patients between June and August 2021 were grouped according to their outcome, immunization status, and the existence of comorbidities. (**A**) Discharged patients without comorbidities; (**B**) Discharged patients with comorbidities; (**C**) Lethal cases without comorbidities; (**D**) Lethal cases with comorbidities. The statistical analysis was performed using the Kruskal–Wallis test for group comparison and Dunn’s test to perform multiple comparisons. The results are shown as the mean ± standard deviation (* *p* < 0.05; *** p* < 0.005; **** p* < 0.001).

**Table 1 ijerph-19-13902-t001:** Discharged patients without comorbidities admitted between June to August 2021 and grouped according to their vaccinal status and the Unified Score for Prioritization (USP).

Vaccinal Status	Discharged Patients without Comorbidities									
	Unified Score of Prioritization (USP)									
	2		3		4		5+		Total	
	*n*	%	*n*	%	*n*	%	*n*	%	*n*	%
Not Iummunized	759	70.47	157	14.58	35	3.25	16	1.49	967	89.79
Partially (D1)	31	2.88	4	0.37	6	0.56	0	0.00	41	3.81
Fully (D2)	29	2.69	15	1.39	10	0.93	15	1.39	69	6.41
**Total**	**819**	**76.04**	**176**	**16.34**	**51**	**4.74**	**31**	**2.88**	**1077**	**100**

**Table 2 ijerph-19-13902-t002:** Discharged patients with comorbidities admitted between June and August 2021 and grouped according to their vaccinal status and the Unified Score for Prioritization (USP).

Vaccinal Status	Discharged Patients with Comorbidities									
	Unified Score of Prioritization (USP)									
	2		3		4		5+		Total	
	*n*	%	*n*	%	*n*	%	*n*	%	*n*	%
Not Iummunized	141	32.87	63	14.69	36	8.39	33	7.69	273	63.64
Partially (D1)	21	4.90	7	1.63	11	2.56	7	1.63	46	10.72
Fully (D2)	27	6.29	20	4.66	21	4.90	42	9.79	110	25.64
**Total**	**189**	**44.06**	**90**	**20.98**	**68**	**15.85**	**82**	**19.11**	**429**	**100**

**Table 3 ijerph-19-13902-t003:** Lethal outcome patients without comorbidities admitted between June and August 2021 and grouped according to their vaccinal status and the Unified Score for Prioritization (USP).

Vaccinal Status	Lethal Outcome Patients without Comorbidities									
	Unified Score of Prioritization (USP)									
	2		3		4		5+		Total	
	*n*	%	*n*	%	*n*	%	*n*	%	*n*	%
Not Iummunized	136	50.75	38	14.18	14	5.22	18	6.72	206	76.87
Partially (D1)	10	3.73	3	1.12	5	1.87	3	1.12	21	7.84
Fully (D2)	12	4.48	6	2.24	8	2.99	15	5.60	41	15.30
**Total**	**158**	**58.96**	**47**	**17.54**	**27**	**10.07**	**36**	**13.43**	**268**	**100**

**Table 4 ijerph-19-13902-t004:** Lethal outcome patients with comorbidities admitted between June and August 2021 and grouped according to their vaccinal status and the Unified Score for Prioritization (USP).

Vaccinal Status	Lethal Outcome Patients with Comorbidities									
	Unified Score of Prioritization (USP)									
	2		3		4		5+		Total	
	*n*	%	*n*	%	*n*	%	*n*	%	*n*	%
Not Iummunized	56	21.96	38	14.90	20	7.84	27	10.59	141	55.29
Partially (D1)	10	3.92	10	3.92	7	2.75	13	5.10	40	15.69
Fully (D2)	6	2.35	16	6.27	19	7.45	33	12.94	74	29.02
**Total**	**72**	**28.24**	**64**	**25.10**	**46**	**18.04**	**73**	**28.63**	**255**	**100**

## Data Availability

The complete dataset presented in this study is available in Zenodo at doi.org/10.5281/zenodo.7249604.

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
