# Peer review of "Effectiveness of COVID-19 Vaccination on Reduction of Hospitalizations and Deaths in Elderly Patients in Rio Grande do Norte, Brazil"

_ijerph, 2022, doi:10.3390/ijerph192113902_

Round 1
Reviewer 1 Report
The authors report hospitalization and death rates of non-vaccinated and vaccinated persons in Brazil between April 2020 and August 2021. The results are interesting and show the efficacy of covid vaccines in reducing severe outcomes. However, there are some issues related to the presentation of the data. The authors mix the cases that took place before vaccinations started with the cases that took place after the start of vaccinations. Furthermore, they seemingly assume that the changes in mortality and morbidity are only due to vaccination and that changes in SARS-CoV-2 variants in different epidemic waves do not play a role for patient outcomes. They also select the study on the population on the basis of the availability of data from the Brazilian data systems RegulaRN and RN+Vaccina and omit other cases, which might lead to a selection bias. The authors should improve their manuscript separating the prevaccination data and presenting it separately. Correspondingly, they should compare the mortality and morbidity data of the cases that took place after the vaccinations started. Furthermore, they should in the discussion evaluate the potential biases in their material (like the differences in the virus variants, effects of a previous covid infection on outcomes of the patients with reinfection and the impact of omitting the cases that did not have data in the Brazilian data system). Finally, they should also present some data on the vaccination coverage of the population that they studied.
Author Response
We appreciate the comments, and, to explore the point-by-point raised, we divided and grouped similar questions, as presented below:
- "The authors mix the cases that took place before vaccinations started with the cases that took place after the start of vaccinations"
and
(4) "The authors should improve their manuscript separating the prevaccination data and presenting it separately. Correspondingly, they should compare the mortality and morbidity data of the cases that took place after the vaccinations started."
Answer: We appreciate the comments and reinforce this differentiation between groups (pre-vaccination vs. post-vaccination; and patients with comorbidities vs. those without comorbidities) in the original version of this manuscript. In fact, we evaluated the impact of the immunization process during the COVID-19 pandemic by analyzing all cases of hospitalized patients from April/2020 to August/2021. The first section of results (3.1 Age and immunization status, but not comorbidities, are associated with the development of moderate and severe COVID-19 cases) was dedicated to exploring the complete database obtained from all patients. Then, in the next section of results (3.2 Vaccination changes the profile of hospitalizations and deaths related to COVID-19), we explored the impact of vaccination on hospitalizations and deaths in two different periods: pre-vaccination (June to August/2020) and post-vaccination (June to August/2021). In both sections, we present data on patients with and without comorbidities (Figure 1 and Figure 3) and the main comorbidities reported during hospitalizations (Figure 2).
(2) "Furthermore, they seemingly assume that the changes in mortality and morbidity are only due to vaccination and that changes in SARS-CoV-2 variants in different epidemic waves do not play a role for patient outcomes."
Answer: In fact, we assume that potentially any of the previously described variants should be found in clinical samples from Brazilian patients in the analyzed period. This fact is due to the recognized flaws in the surveillance system of viral genomes in circulation and the absence of effective barriers to prevent the entry of patients infected in other countries into Brazil (e.g., restriction of flights in critical periods and large-scale testing requirements for all flights). Despite this limitation, all viral strains identified in this period (including alpha, beta, gamma, and delta) were associated with an increase in transmissibility worldwide but not directly associated with a significant increase in lethality (except when there was a saturation of healthcare services and not directly caused by the increase of viral pathogenesis). Recently, Brizzi and colleagues evidenced that the geographic and temporal fluctuations in Brazil's COVID-19 in-hospital lethality rates between January 2020 and July 2021 were primarily associated with geographic inequities and shortages in healthcare capacity [1], which is also corroborated by the RegulaRN public database [2]. Taken together, our data and the data from worldwide lead us to the unequivocal conclusion that vaccination was primarily responsible for reducing new cases and deaths, despite new viral strains in this period.
(3) "They also select the study on the population on the basis of the availability of data from the Brazilian data systems RegulaRN and RN+Vaccina and omit other cases, which might lead to a selection bias"
Answer: We appreciate the comment, but here we need to clarify some aspects of this study's primary data source. All data from COVID-19 patients were obtained from the official records from the Secretary of Public Health of Rio Grande do Norte (SESAP/RN), the local authority for managing COVID-19 cases in Rio Grande do Norte state, which is subordinated to the Brazilian Ministry of Health. As mentioned in the manuscript (lines 127 to 133), we exclude from this analysis the data from patients with an unconfirmed COVID-19 diagnosis (e.g., suspected, discarded, and unreported cases), patients admitted directly by the state's health care network (which frequently does not present complete medical records in the RegulaRN system) and those patients with unknown outcomes (discharge or death) at the end of the hospitalization period. These criteria were adopted precisely to avoid bias during the data analysis. In addition, we reinforce that the complete anonymized data presented in this study will be available in Zenodo at doi.org/10.5281/zenodo.6560919.
(5) "Furthermore, they should in the discussion evaluate the potential biases in their material (like the differences in the virus variants, effects of a previous covid infection on outcomes of the patients with reinfection and the impact of omitting the cases that did not have data in the Brazilian data system)"
Answer: We appreciate the suggestion and have included in the discussion of the updated version of the manuscript the impact of immunization even during the spread of different viral strains, as previously evidenced here. Regarding the COVID-19 cases not registered in the official records of the Brazilian Ministry of Health, we understand that sub-notification of COVID-19 cases may occur in several countries (according to their test capacity over the pandemic). Since this situation is beyond our control, exploring the reinfection of COVID-19 cases particularly is not the goal of this study, and this aspect does not invalidate our findings presented in this manuscript.
(6) "Finally, they should also present some data on the vaccination coverage of the population that they studied"
Answer: We appreciate the comment and have added this information in the updated manuscript version. According to recent estimates from the Brazilian Institute of Geography and Statistics (IBGE), the Rio Grande do Norte state presented a 3,560,903 population [3]. Considering the end of August 2021, 62.82% of this population was vaccinated with at least one dose, while 24.90% was immunized with two doses. Considering only the elderly population (479,536 habitants), at the end of August 2021, 98% of this population was vaccinated with one dose, and 91,94% of the elderly were immunized with two doses. This alteration is highlighted in the updated manuscript version.
REFERENCES
- Brizzi, A.; Whittaker, C.; Servo, L.M.S.; Hawryluk, I.; Prete, C.A.; de Souza, W.M.; Aguiar, R.S.; Araujo, L.J.T.; Bastos, L.S.; Blenkinsop, A.; et al. Spatial and Temporal Fluctuations in COVID-19 Fatality Rates in Brazilian Hospitals. Nat Med 2022, 28, 1476–1485, doi:10.1038/s41591-022-01807-1.
- LAIS/UFRN Plataforma RegulaRN - Sala de Situação Pública Available online: https://regulacao.saude.rn.gov.br/sala-situacao/sala_publica/ (accessed on 19 April 2022).
- IBGE Censo Do Rio Grande Do Norte, Brasil Available online: https://cidades.ibge.gov.br/brasil/rn/panorama (accessed on 5 May 2022).
- Clemens, S.A.C.; Folegatti, P.M.; Emary, K.R.W.; Weckx, L.Y.; Ratcliff, J.; Bibi, S.; de Almeida Mendes, A.V.; Milan, E.P.; Pittella, A.; Schwarzbold, A. v.; et al. Efficacy of ChAdOx1 NCoV-19 (AZD1222) Vaccine against SARS-CoV-2 Lineages Circulating in Brazil. Nat Commun 2021, 12, 5861, doi:10.1038/s41467-021-25982-w.
- Cerqueira-Silva, T.; Andrews, J.R.; Boaventura, V.S.; Ranzani, O.T.; de Araújo Oliveira, V.; Paixão, E.S.; Júnior, J.B.; Machado, T.M.; Hitchings, M.D.T.; Dorion, M.; et al. Effectiveness of CoronaVac, ChAdOx1 NCoV-19, BNT162b2, and Ad26.COV2.S among Individuals with Previous SARS-CoV-2 Infection in Brazil: A Test-Negative, Case-Control Study. Lancet Infect Dis 2022, 22, 791–801, doi:10.1016/S1473-3099(22)00140-2.

Reviewer 2 Report
General
This is a good epidemiological work covering outcomes of vaccination in a state in Brazil April 2020 to August 2021.
Under introduction
The authors review the state of the epidemic at the aforementioned time period so as to provide the reader with the specific context. Having the data available to all is an advantage.
Under results
The Figures are too difficult to apprehend. The authors need to help the readers to see it's a 2x2 or 2x3 table (as if). It's not easy to apprehend what the notes under each panel and the colors denote.
Under discussion
It is unclear why the authors focus on the 4 platforms, which were not analyzed in the current study. It may be of interest, but there is no data in this analysis on another set-up, e.g., one in which only 3, or 2 platforms or one platform were used. In the absence of data and analysis on this aspect, it should not be dwelt on.
Excluding individuals who were transferred should be listed under study's weaknesses. Were people transferred due to complications?
Misc:
1. Use abbreviations only after explaining them for the first time (e.g. in the abstract)
2. Consider the verb "applied" to vaccination; does not seen appropriate to me, though not a native speaker.
Author Response
Under results:
The Figures are too difficult to apprehend. The authors need to help the readers to see it's a 2x2 or 2x3 table (as if). It's not easy to apprehend what the notes under each panel and the colors denote.
Answer: We appreciate the comments and suggestions. Since we have many data and to avoid splitting the information into several figures, we grouped the data in tables and colored figures with legends below the respective groups. However, we welcome new editorial strategies to present the data differently.
Under discussion:
It is unclear why the authors focus on the 4 platforms, which were not analyzed in the current study. It may be of interest, but there is no data in this analysis on another set-up, e.g., one in which only 3, or 2 platforms or one platform were used. In the absence of data and analysis on this aspect, it should not be dwelt on. Excluding individuals who were transferred should be listed under study's weaknesses. Were people transferred due to complications?
Answer: We welcome the comments and would like to explore some points raised. Along with the immunization process, the Brazilian Ministry of Health made four different vaccine platforms available for the general population, resulting in an occasional mix-and-match strategy using the BNT162b2 (Pfizer/BioNTech), CoronaVac (Sinovac Biotech), ChAdOx1 nCoV-19 (Oxford/AstraZeneca), and Ad26.COV2.S (Jassen). Thus, this study aimed not to compare the efficacy of different immunizers (all approved by WHO and widely explored elsewhere) but to assess the impact on reducing hospitalizations and deaths in elderly patients right after administering at least one dose. To explore this question, we analyze data from two independent platforms, RegulaRN (hospitalizations and deaths associated with COVID-19 cases) and RN+Vacina (that records the immunization data from all patients). These platforms are the official source of data from the Secretary of Public Health of Rio Grande do Norte (SESAP/RN), the local authority for managing COVID-19 cases in Rio Grande do Norte state, which is subordinated to the Brazilian Ministry of Health. The complete anonymized dataset presented in this study will be available in Zenodo at doi.org/10.5281/zenodo.6560919.
The reason for the exclusion of some individuals in this study was due to unconfirmed COVID-19 diagnosis (e.g., suspected, discarded, and unreported cases), patients admitted directly by the state's health care network (which frequently does not present complete medical records in the RegulaRN system) and those patients with unknown outcomes (discharge or death) at the end of the hospitalization period. In addition, all patients are most likely transferred to a less specialized hospital due to improved clinical conditions. However, as we were unsure about the outcome, we excluded all data from these patients to avoid bias during the analyses.
Misc:
- Use abbreviations only after explaining them for the first time (e.g. in the abstract);
and
- Consider the verb "applied" to vaccination; does not seen appropriate to me, though not a native speaker.
Answer: We appreciate the suggestions and inform you that we have accepted all recommendations. The respective alterations are highlighted in the updated version of the manuscript.

Reviewer 3 Report
This is an important study, among several others, confirming the effectiveness of COVID-19 vaccination in reducing mortality and hospitalization in older adults.
I have a couple of suggestions below:
1. In line 75, please mention if the study aimed to look at the effectiveness of the COVID-19 vaccine in the elderly. as the current description seems like for the overall population.
2. Line 150 to 153 states that the participant who received the vaccine from June to Aug 201 were included, and these were preferably 60 years or older persons. I understand this is the reason study is titled as, "assessing the effectiveness in elderly patients." Hence this information should fit better in section 2.1 and should be presented in a more transparent way.
Author Response
- In line 75, please mention if the study aimed to look at the effectiveness of the COVID-19 vaccine in the elderly. as the current description seems like for the overall population.
Answer: We welcome this suggestion and inform you that we have accepted the recommendation. This alteration is highlighted in the updated manuscript version.
- Line 150 to 153 states that the participant who received the vaccine from June to Aug 201 were included, and these were preferably 60 years or older persons. I understand this is the reason study is titled as, "assessing the effectiveness in elderly patients." Hence this information should fit better in section 2.1 and should be presented in a more transparent way.
Answer: We fully agree with the suggestion and include this information in section 2.1 (Caseload and data acquisition) of the materials and methods. This alteration is highlighted in the updated manuscript version.

Reviewer 4 Report
In this manuscript, “Effectiveness of COVID-19 vaccination on reduction of hospitalizations and deaths in elderly patients in Rio Grande do Norte, Brazil” by Sales-Moioli et al. evaluates the impact of the immunization process on moderate or severe COVID-19 cases and explore how those patients performed considering their vaccination status at the time of admission. Although, authors had provided some statistical results, this work did not separate the effect of vaccine from BNT162b2 (Pfizer/BioNTech), CoronaVac 63 (Sinovac Biotech), ChAdOx1 nCoV-19 (Oxford/AstraZeneca), or Ad26.COV2.S (Jassen). Therefore, I would suggest authors may take at least a major revision before publication. Here are the comments and suggestions:
1. The current results and conclusions agreed with other reports; however, these vaccines had various types. Could authors comment about the ratio and effective of each type of vaccines?
Author Response
Answer: We appreciate the comments and the question. In fact, despite all four immunizers evaluated in this study being approved by WHO and widely administered worldwide, the comparison between the immunizers may represent a relevant aspect for future studies that may help to improve the vaccination process and the development of new vaccines. Along with the immunization process, the Brazilian Ministry of Health made four different vaccine platforms available for the general population, resulting in an occasional mix-and-match strategy using the BNT162b2 (Pfizer/BioNTech), CoronaVac (Sinovac Biotech), ChAdOx1 nCoV-19 (Oxford/AstraZeneca), and Ad26.COV2.S (Jassen). Recently, Clemens and colleagues evidenced the efficacy of the ChAdOx1 nCoV-19 (Oxford/AstraZeneca) vaccine against the viral strains circulating in Brazil. In parallel, Cerqueira-Silva and colleagues explored the effectiveness of these immunizers in individuals with previous SARS-CoV-2 infection [4,5]. However, considering this direct comparison was not our primary goal in this study, we did not perform this analysis and focused on the impact of reducing hospitalizations and deaths in elderly patients after immunization. Nevertheless, the complete anonymized dataset presented in this study will be available in Zenodo at doi.org/10.5281/zenodo.6560919 and can be helpful for developing new studies.
REFERENCES
- Brizzi, A.; Whittaker, C.; Servo, L.M.S.; Hawryluk, I.; Prete, C.A.; de Souza, W.M.; Aguiar, R.S.; Araujo, L.J.T.; Bastos, L.S.; Blenkinsop, A.; et al. Spatial and Temporal Fluctuations in COVID-19 Fatality Rates in Brazilian Hospitals. Nat Med 2022, 28, 1476–1485, doi:10.1038/s41591-022-01807-1.
- LAIS/UFRN Plataforma RegulaRN - Sala de Situação Pública Available online: https://regulacao.saude.rn.gov.br/sala-situacao/sala_publica/ (accessed on 19 April 2022).
- IBGE Censo Do Rio Grande Do Norte, Brasil Available online: https://cidades.ibge.gov.br/brasil/rn/panorama (accessed on 5 May 2022).
- Clemens, S.A.C.; Folegatti, P.M.; Emary, K.R.W.; Weckx, L.Y.; Ratcliff, J.; Bibi, S.; de Almeida Mendes, A.V.; Milan, E.P.; Pittella, A.; Schwarzbold, A. v.; et al. Efficacy of ChAdOx1 NCoV-19 (AZD1222) Vaccine against SARS-CoV-2 Lineages Circulating in Brazil. Nat Commun 2021, 12, 5861, doi:10.1038/s41467-021-25982-w.
- Cerqueira-Silva, T.; Andrews, J.R.; Boaventura, V.S.; Ranzani, O.T.; de Araújo Oliveira, V.; Paixão, E.S.; Júnior, J.B.; Machado, T.M.; Hitchings, M.D.T.; Dorion, M.; et al. Effectiveness of CoronaVac, ChAdOx1 NCoV-19, BNT162b2, and Ad26.COV2.S among Individuals with Previous SARS-CoV-2 Infection in Brazil: A Test-Negative, Case-Control Study. Lancet Infect Dis 2022, 22, 791–801, doi:10.1016/S1473-3099(22)00140-2.

Round 2
Reviewer 4 Report
It seems that this work can not tell the impact of these four vaccines on reducing hospitalizations and deaths in elderly patients after immunization.
Author Response
We appreciate the comment, but, as previously mentioned in the first review report, the direct comparison between immunizers in reducing hospitalization and deaths was not part of our goal in the present paper. This analysis has not been previously performed to avoid a predilection for a specific immunizing agent based on its performance in the evaluated population. However, to address this question, we analyzed the anonymized dataset available in Zenodo (doi.org/10.5281/zenodo.6560919) and summarized this data in the following tables.
As presented in Supplementary Table 1, the Not immunized patients represented 77.97% of all hospitalizations and 66.16% of all deaths observed in this period (June to August 2021), when all immunizers were widely available to the population. In parallel, all patients who received shots from Pfizer (n=640,073) or Janssen (n= 56,087) immunizers were not hospitalized or had a lethal outcome related to Covid-19 in this period. Not immunized patients were responsible for 77.97% (n=1,582) of hospitalizations and 66.16% (n=346) of all deaths observed in this period.
Supplementary Table 1. Vaccines administered, hospitalizations and deaths related to Covid-19 grouped according to manufacturing sources observed in the adult population (18+ years old) between June to August 2021 in the Rio Grande do Norte, Brazil.
|
Immunizers |
Vaccines administered |
Hospitalizations |
Deaths |
|||
|
n |
% |
n |
% |
n |
% |
|
|
BNT162b2 (Pfizer/BioNTech) |
640,073 |
32.80 |
- |
- |
- |
- |
|
ChAdOx1 nCoV-19 (Oxford/AstraZeneca) |
961,143 |
49.26 |
152 |
7.49 |
60 |
11.47 |
|
CoronaVac (Sinovac Biotech/Butantan) |
293,854 |
15.07 |
295 |
14.54 |
117 |
22.37 |
|
Ad26.COV2.S (Janssen) |
56,087 |
2.87 |
- |
- |
- |
- |
|
Not immunized |
- |
- |
1,582 |
77.97 |
346 |
66.16 |
|
Total |
1,951,157 |
100 |
2,029 |
100 |
523 |
100 |
Considering only the elderly patients immunized in this period (Supplementary Table 2), we observed that Oxford/AstraZeneca was administered in 91.46% (n=155,822) of all elderly immunized but was associated with 23.5% (n=145) of all hospitalizations and 21.9% (n=57) of all deaths observed in this period. All patients who received shots from Pfizer (n=6,078) or Janssen (n=776) were not hospitalized or had a lethal outcome related to Covid-19 in this period. Additionally, 4.52% of all elderly immunized patients in this period (n=7700) received a CoronaVac shot (first or second dose). This group represented 47.65% (n=294) of all hospitalizations and 45% (n=117) deaths observed in elderly patients in this period.
Supplementary Table 2. Vaccines administered, hospitalizations and deaths related to Covid-19 grouped according to manufacturing sources observed in the elderly population (60+ years old) between June to August 2021 in the Rio Grande do Norte, Brazil.
|
Immunizers |
Vaccines administered |
Hospitalizations |
Deaths |
|||
|
n |
% |
n |
% |
n |
% |
|
|
BNT162b2 (Pfizer/BioNTech) |
6,078 |
3.57 |
- |
- |
- |
- |
|
ChAdOx1 nCoV-19 (Oxford/AstraZeneca) |
155,822 |
91.46 |
145 |
23.5 |
57 |
21.9 |
|
CoronaVac (Sinovac Biotech/Butantan) |
7,700 |
4.52 |
294 |
47.65 |
117 |
45 |
|
Ad26.COV2.S (Janssen) |
776 |
0.45 |
- |
- |
- |
- |
|
Not immunized |
- |
- |
178 |
28.85 |
86 |
33.1 |
|
Total |
170,376 |
100 |
617 |
100 |
260 |
100 |
Nevertheless, it is essential to consider that, besides being an inactivated virus vaccine, CoronaVac was initially administered to healthcare professionals and older patients (80+ years old), which may be associated with a higher number of comorbidities or risk factors associated with lethal outcomes during Covid-19 infection. These questions are pertinent and must be explored in further studies, as this paper can't address all relevant points in a single study.
